# Skull Base Reconstruction by Subsite after Sinonasal Malignancy Resection

**DOI:** 10.3390/cancers16020242

**Published:** 2024-01-05

**Authors:** Kristen Kraimer, Mathew Geltzeiler

**Affiliations:** Department of Otolaryngology—Head and Neck Surgery, Oregon Health & Science University, Portland, OR 97239, USA

**Keywords:** sinonasal malignancy, skull base reconstruction, nasoseptal flap, sinonasal squamous cell carcinoma, cerebrospinal fluid leak repair

## Abstract

**Simple Summary:**

Sinonasal malignancies involving the skull base often involve extensive resection and complex reconstruction. Reconstruction efforts must aim to prevent postoperative cerebrospinal fluid leak, pneumocephalus, and infection, in addition to restoring paranasal sinus function. Reconstruction after sinonasal malignancy resection is particularly complex, as many typical skull base reconstructive options are limited due to the extent of resection and margins. The nasoseptal flap is a workhorse of skull base reconstruction but is not consistently available, especially in reoperations and settings of advanced locoregional malignancy. This review aims to discuss the particular reconstructive challenges and analyze the current literature on reconstructive practices in all sinonasal subsites.

**Abstract:**

Reconstruction after the resection of sinonasal malignancies is complex and primarily dependent on the defect size and location. While the reconstructive paradigm for sellar mass resection is well delineated, the challenges associated with reconstruction after sinonasal malignancy resection are less well described. This narrative review will address the goals of reconstruction after both endonasal endoscopic and open sinonasal malignancy resection and reconstructive options specific to these subsites. The goals of reconstruction include repairing cerebrospinal fluid leaks, restoring sinonasal function, providing a nasal airway, and optimizing the patient’s quality of life. These goals are often complicated by the anatomic nuances of each involved sinus. In this review, we will discuss the methods of reconstruction specific to each sinonasal subsite and describe the factors that guide choosing the optimal reconstructive technique.

## 1. Introduction

Reconstruction after sinonasal malignancies is complex, and techniques are highly variable among the sinonasal subsites and extent of the tumor. Similar to general reconstructive paradigms, the goal is to restore form and function after the resection of sinonasal malignancies. This includes the reconstruction of the skull base, orbit, nasal airway, and palate. Particular attention is paid to skull base reconstruction to prevent postoperative cerebrospinal fluid (CSF) leaks, pneumocephalus, and meningitis. Historically, large skull base defects were associated with high rates, up to 20%, of postoperative CSF leaks, meningitis, and poor survival [1]. Through significant improvement and innovation in skull base reconstruction, the rates of these devastating complications are lower [1,2].

Sinonasal malignancies make up about 5% of all head and neck malignancies, and the most common histologic diagnosis is squamous cell carcinoma [3]. Other sinonasal malignancy diagnoses include sinonasal adenocarcinoma, sinonasal neuroendocrine carcinoma, sinonasal undifferentiated carcinoma, minor salivary gland tumors, mucosal melanoma, and esthesioneuroblastoma. Among these, the five-year overall survival varies between 22 and 70% [3]. The most common locations for sinonasal malignancies are the nasal cavity, maxillary sinus, and ethmoid sinuses. In addition to poor survival metrics, patients with head and neck cancer are known to have significant declines in quality of life and body image after surgical treatment due to the effects on cosmesis, the nasal airway, voicing, communication, swallowing, and aesthetics [4,5]. In cases of sinonasal malignancies, adjuvant radiation can also lead to significant crusting and nasal obstruction that further affect the quality of life in this patient population. This is important to consider in reconstructive efforts and has been an increasing area of research as we refine our surgical and reconstructive techniques. The focus of this narrative review is on the principles of skull base reconstruction in the setting of sinonasal malignancies. 

## 2. Reconstructive Goals

The goals for the repair of the skull base after malignancy resection are to repair cerebrospinal fluid (CSF) leaks, separate the intracranial contents from paranasal sinuses, and restore paranasal sinus function. CSF leak repair technique choice is affected by defect size, location, flow volume (high vs. low), and patient factors like age, body mass index, and the need for adjuvant therapy. High-flow leaks are typically defined as a dural defect greater than 1 square centimeter and/or entering a CSF cistern (i.e., the suprasellar or prepontine cistern) [6]. Repair options include grafts and vascularized flaps. Grafts include turbinate mucosa, septal mucosa, nasal mucosa, abdominal or anterolateral thigh fascia, abdominal fat, and alloplastic products. Vascularized flaps, particularly the nasoseptal flap, have become a mainstay in skull base reconstruction to prevent CSF leaks and risk for meningitis. Other vascularized regional flaps include pericranium, temporoparietal fascia, facial artery buccinator, and palatal flaps. Finally, free tissue transfer can be considered for large defects and, typically, radial forearm or latissimus free tissue flaps are used in these scenarios. Multiple studies have shown that smaller defects of 1 cm^2^ or less can be successfully reconstructed with grafts alone, while large defects with high flows have significantly improved repair rates when vascularized tissue is used [7]. Additionally, multi-layer reconstruction is critical for patients with very large defects [8,9].

Paranasal sinus restoration is also important to consider during skull base reconstruction to improve patient postoperative quality of life, prevent infection, and prevent mucocele formation. Ideally, tissue preservation should be employed, when possible, and sinonasal mucosa should not be stripped unless necessary for oncologic purposes. Special care must be taken near the frontal recess, as post-op frontal sinus mucoceles are a common occurrence, especially after the treatment of sinonasal malignancies and adjuvant radiation therapy. If it is impossible to maintain sinus patency, the mucosa of the affected sinus should be stripped completely during the index surgery to prevent post-op trapped mucosa and mucocele formation. Additionally, regular postoperative debridement procedures after endoscopic sinus surgery have been shown to have a significant benefit to postoperative healing and long-term symptom improvement [10,11,12]. The optimal intervals for office debridements have been heavily discussed in the literature and are dependent on the degree of surgery, patient factors, and findings during subsequent debridement procedures. After endoscopic sinus surgery, debridement is typically performed one week after surgery, but this has not been clearly studied in patients who underwent skull base reconstruction for malignancies [10]. However, debridements should be performed until the patient is no longer crusting and/or is symptomatically stable.

The restoration of nasal appearance and airway function is an additional consideration for patients with sinonasal malignancies. Some of these lesions require partial or total rhinectomy and partial or total maxillectomies, in addition to orbitotomies and exenterations. Surgical teams should be comfortable with the wide range of reconstructive techniques for nasal and facial reconstruction, including the full complement of local and regional flaps plus free tissue transfer. Prosthetists are also important team members who can create nasal prostheses and maxillary and facial obturators. 

An emerging consideration in complex skull base reconstruction is virtual surgical planning. This has been heavily described in the bony mandibular and maxillary reconstruction of the head and neck but rarely described for anterior skull base reconstruction. Kayastha et al. described the use of a 3D model for planning a nasoseptal flap in order to improve mucosal preservation [13]. They used a 3D model based on patients’ preoperative CT scans to measure the optimal width of the nasoseptal flap to adequately reconstruct the anticipated anterior skull base defect while minimizing excess length. This was performed in three patients and was noted to be a viable reconstructive tool. Another group recently analyzed a group of patients undergoing midface reconstruction with subscapular system free tissue transfer with and without virtual surgical planning [14]. They found that utilizing virtual surgical planning resulted in a significantly higher number of successfully reconstructed midface subunits, successful bony contact between segments, and percent of segments in anatomic position [14]. This is a tool to be considered, especially in free flap reconstruction, after sinonasal malignancy resection, but there is a significant cost associated with these systems. Future studies are needed to better delineate the potential long-term benefits for patients, cost savings, and true cost of utilizing these technologies in reconstructive efforts.

## 3. Skull Base Reconstructive Ladder

Skull base reconstruction techniques are dependent on the size of the defects and the presence of high-flow CSF leaks. Not unlike reconstruction algorithms for other sites, skull base reconstruction can be characterized as a reconstructive ladder. For small defects of less than 1 cm^2^ and low-flow CSF leaks, mucosal-free grafts are an acceptable strategy [15]. Free mucosal grafts can be harvested from the inferior or middle turbinate, nasal floor, nasal sidewall, or septum. These have been shown to be sufficient for small defects of less than 1 cm^2^, with no CSF leak or a low-flow CSF leak [8,15]. Free mucosal grafts can be a beneficial tool as they can be harvested at the primary surgical site with a range of harvest sites within the nasal cavity and can be used as elements in a multi-layer closure. However, free mucosal grafts are not sufficient for large defects or high-flow CSF leaks. Free grafts can also be utilized to hasten remucosalization in areas with exposed cartilage and bone. Table 1 provides a summary of the various reconstructive options that are discussed in further detail below.

### 3.1. Multi-Layer Closure

Multi-layer closure techniques are an essential component of skull base reconstruction and have decreased postoperative CSF leak rates to between 5–10% [2]. In patients with large defects and low-flow CSF leaks or patients with high-flow CSF leaks, multi-layer repair is recommended [8]. This often consists of subdural inlay grafts, onlay grafts (non-mucosalized), and a vascularized flap. The types of inlay grafts described include synthetic collagen substitutes, an acellular dermal matrix, temporalis fascia, and fascia lata. Fascia lata or alloplastic material is commonly used for onlay grafts, and particular attention is given to adequately overlapping the graft onto the surrounding bone, removing excess adipose tissue, and ensuring that the surrounding bone is stripped of any mucosa [9]. Figure 1 demonstrates an example of anterior skull base reconstruction with synthetic dura, fascia lata, and pedicled nasoseptal flap.

### 3.2. Fascia Lata

Fascia lata is a non-pedicled free graft that can be used for inlay, onlay, and button grafts for multi-layer skull base reconstruction [16]. This is often performed in conjunction with a nasoseptal flap and offers the benefit of minimal donor site morbidity. However, it is harvested from the thigh, so another sterile surgical field is required. It does have the benefit of potential simultaneous harvest during the endoscopic endonasal portion of the procedure.

### 3.3. Nasoseptal Flap

The nasoseptal flap was first described in 2006 and has significantly changed the practice of skull base surgery by decreasing rates of postoperative CSF leaks and meningitis [2,17]. This provides a large, vascularized graft to repair skull base defects and is considered the workhorse of skull base reconstruction because it can be used to reconstruct all regions of the central skull base. Among patients with high-flow intraoperative CSF leaks, nasoseptal flap reconstruction typically results in a postoperative CSF leak of 5% [7]. Compared to reconstruction with free grafts for large dural defects, nasoseptal flaps demonstrate a significant benefit in reducing postoperative CSF leaks [2]. This flap is also versatile and can be tailored for specific defects. The standard flap is raised along the septum with the inferior incision at the most inferior aspect of the septum at the nasal floor. However, an extended nasoseptal flap can also be harvested, which includes the nasal floor mucosa to the inferior meatus to increase the area of the flap [18]. Figure 2 demonstrates a well-healed nasoseptal flap on postoperative T1 post-contrast MRI images that were used for skull base reconstruction after pituitary mass resection. MRI images can be helpful in the postoperative nasoseptal flap assessment if there is a question of flap viability and if it is not clinically feasible to perform a debridement for direct inspection. The nasal flap is also versatile and can reconstruct defects from the lower portion of the posterior table of the frontal sinus to the clivus and foramen magnum. Despite the inherent reconstructive benefits of the nasoseptal flap, there is associated donor site morbidities, including nasal crusting, drainage, decreased olfaction, septal perforation, and nasal dorsum collapse [19,20]. The reverse nasoseptal flap is a technique that has been recently described to decrease the morbidity of the NSF with reports of decreased crusting and patient reported nasal deformities [20,21]. However, this is accompanied by the limitation of preventing use of the contralateral septal mucosa for another nasoseptal flap. Another option to decrease crusting after nasoseptal flap harvest is securing a free mucosal graft onto the exposed septum to facilitate remucosalization.

### 3.4. Pericranial Flap

The use of nasoseptal local flaps can be limited, especially in the setting of malignancy, if the septal mucosal is involved or sacrificed during the resection or in the setting of prior surgery. In that case, alternative local flaps and regional or free flaps can be considered. Pericranial flaps have been heavily described for many reconstructive efforts in the head and neck and after bifrontal craniotomy [22]. If a frontal craniotomy is utilized for the resection of the malignancy, the flap can be inserted in the traditional fashion. However, the blood supply to pericranial flaps may be compromised during the resection for frontal sinus tumors and thus may limit its use if at least one supraorbital vessel bundle is not intact [23]. More recently, the extracranial pericranial flap has been adapted to accommodate anterior fossa defects created purely endoscopically [24,25]. This is performed by raising the pericranial flap via a traditional bicoronal incision and then tunneling through a small skin incision and frontal osteotomy before insetting it endoscopically at the site of the defect [24]. These flaps provide robust vascularized tissue that can reach from the posterior table of the frontal sinus back past the sella and clivus. Multiple retrospective reviews of nine patients who underwent reconstruction with a pericranial flap during endoscopic endonasal approaches to the anterior skull base demonstrated it to be a reliable and effective reconstructive tool [24,25,26]. The pericranial flap has the limitations of adding another operative site, and incision for harvest and donor site morbidities include pain, numbness, alopecia, and poor cosmesis.

### 3.5. Additional Locoregional Pedicled Flaps

In cases when a nasoseptal flap is unavailable due to the extent of resection or prior surgery, several other vascularized flaps have been described for skull base reconstruction. Tempoparietal fascial (TPF) pedicled flaps have also been described for anterior skull base reconstruction. TPF flaps are based on the superficial temporal artery and are useful due to their consistent anatomy for harvest and the large surface area available for reconstruction [27,28]. It is useful for expanded sphenoid or clival reconstruction not typically used for anterior skull base reconstruction due to the limited arc of rotation [27]. Donor site morbidities include alopecia, injury to the temporal branch of the facial nerve, and internal maxillary artery injury [28].

Inferior turbinate flaps and lateral nasal wall flaps are another option, although they are not commonly used. These are pedicled on the inferior turbinate artery and are an option for a vascularized flap reconstruction of the sella [27]. Additionally, palatal flaps have been modified from their use in cleft palate reconstruction to be used for the reconstruction of the planum, sella, and clivus [29]. These are based on the descending palatine artery and can yield a large flap, up to 18 cm^2^, with a 3 cm pedicle [29]. The relatively long pedicle length is advantageous to increase options for vessel anastomoses and avoid additional vein grafts. A facial artery buccinator flap has also been described for the reconstruction of the anterior skull base and planum sphenoidal and is based on the reverse flow from the angular artery [30].

### 3.6. Free Tissue Transfer

Free tissue transfer is an important technique to consider in anterior skull base reconstruction when locoregional pedicled flaps are unavailable or significant soft tissue bulk is needed for adequate reconstruction. Several donor sites have been described and include the radial forearm, anterolateral thigh, rectus abdominis, serratus anterior, latissimus dorsi, and fibula [31,32]. The key consideration is filling the soft tissue defect and closing the CSF leak. Weber et al. described an algorithm for donor site selection based on the size of the defect. They suggested that small defects, <40 cc, can be reconstructed with the radial forearm, scapula, serratus, or ulna flaps, while larger defects, such as those after maxillectomy and orbital exenteration, may require an anterolateral thigh, rectus abdominis, or latissimus dorsi flap [31]. Figure 3 demonstrates preoperative and postoperative MRI images of recurrent adenoid cystic carcinoma and subsequent skull base reconstruction with an anterolateral thigh free flap. Fibula and subscapular system flaps have the benefit of osseous reconstruction to reconstruct the bony buttresses of the face for function and cosmesis [8,32]. The most common vessels to sue for anastomosis are the facial artery and vein. The superficial temporal artery and vein have been described as the ideal recipient vessels for free tissue transfer due to their anatomic location and vessel caliber [31]. However, these vessels may be unavailable due to resection, radiation injury, or prior surgery. Free tissue transfer also has the advantage of strong vascularization to withstand postoperative radiation compared to locoregional flaps for skull base reconstruction [8]. This is important to consider, especially in the context of reconstruction after sinonasal malignancy resection, because many patients will go on to require postoperative radiation therapy.

Lumbar drains are also an essential consideration in skull base reconstruction. A randomized control trial assessing the potential benefit of perioperative lumbar drains in skull base reconstruction concluded that the use of a lumbar drain among patients with high-flow CSF leaks and nasoseptal flap reconstruction resulted in significantly lower rates of postoperative CSF leaks [6]. Inclusion criteria to define a high-flow leaks included a dural defect greater than one square centimeter, extensive arachnoid dissection, and/or dissection into a ventricle or cistern [6].

## 4. Maxillary Sinus

Reconstruction of primary maxillary sinus malignancies is dependent on the extent of the mass and involvement of surrounding structures. In situations when the mass is confined to the sinus and resection is achieved endoscopically, it is important to ensure an adequate outflow pathway after resection to include the natural ostium. Postoperative debridements are important to maintain patency and restore sinonasal function [10,11,12]. Further study in patients undergoing endoscopic mass excision for malignancy, rather than endoscopic sinus surgery for chronic sinusitis, is needed to determine the optimal interval for debridements and irrigation recommendations.

The extent of maxillectomy for these tumors guides the reconstructive needs and options. When an endoscopic medial maxillectomy is performed, typically the wound bed is left to granulate and remucosalize. These patients require frequent clinic debridements to prevent obstructive crusting and synechiae. A nasal floor mucosal graft or a medially based nasal floor flap can also be used in these patients, especially if a large medial maxillectomy defect is created, to aid in faster remucosalization.

### 4.1. Reconstruction after Partial Maxillectomy

Reconstruction after infrastructure maxillectomy, with the preservation of the orbit, requires additional anatomic considerations. This review focuses on skull base reconstruction, but when the palate and skull base are both involved, significant soft tissue needs to be replaced. Palatal reconstruction is required to maintain oronasal separation. This can be achieved with obturator use, local flaps, and free tissue transfer. An obturator provides the benefits of faster operative time and continued ability for visual inspection of the tumor site during postoperative surveillance visits [33]. Obturators also allow the other locoregional flaps to be used for the skull base reconstruction so that a free flap is not required. The challenge with an obturator, in addition to the expensive cost for the patient, is that this is removed and replaced daily. The patient will continue to have hypernasal speech when the obturator is out, such as at nighttime when they are sleeping. Additionally, successful use requires ample dexterity, which may not be possible for all patients, especially elderly patients with advanced arthritis. Finally, obturator use may be challenging particularly in sinonasal malignancy patients due to the frequent use of adjuvant radiation and common side effects of postoperative xerostomia [33].

A temporalis muscle pedicled graft can be considered for small, less than 50% palate, defects and provides the benefit of harvest within the same surgical site [33]. The disadvantages include relatively small available tissue bulk, which may be more problematic in sinonasal malignancy patients who are expected to undergo adjuvant radiation therapy.

Free tissue transfer options include myocutaneous, fasciocutaneous, and osteocutaneous flaps. A fasciocutaneous radial forearm flap is a good option to obliterate the maxillary sinus and close small palatal defects, less than 50% of the transverse palate, but will not provide bony support for future dental rehabilitation [34]. Rectus abdominus and serratus anterior myocutaneous flaps have also been described for small infrastructure reconstructions, and the choice of the donor site is highly variable among microsurgeons [33].

### 4.2. Reconstruction after Total Maxillectomy

Total maxillectomy requires consideration of orbital reconstruction, in addition to the above infrastructure maxillectomy considerations. Compared to infrastructure reconstructions, total maxillectomy reconstruction with or without orbital exenteration requires a larger volume of tissue and typically requires free flap reconstruction. Typically, an obturator does not provide sufficient support for the orbit, so definitive surgical reconstruction is needed. Careful consideration of the orbital floor is important because inadequate orbital floor reconstruction can lead to hypoglobus, enophthalmos, diplopia, and bothersome cosmesis. Several reconstructive strategies have been discussed in the literature and often include a combination of soft tissue rearrangement, non-vascularized bone grafts, bony-free tissue transfer, and alloplastic implants [34]. Non-vascularized bone grafts, such as scapular or iliac crest, and vascularized bone-free tissue transfers have been found to be acceptable methods of orbital floor reconstruction. However, bony-free flaps have been found to have decreased rates of exposure and infection [33]. Titanium mesh and porous polyethylene implants have been described in orbital floor reconstruction but carry the risk of extrusion and infection, especially in the setting of adjuvant radiation, which is common among the sinonasal malignancy population [33,35,36]. For patients who may require adjuvant radiation, alloderm is a viable alternative for the reconstruction of the orbital floor [37]. Other authors describe that if the periorbita is intact, typically no bony orbital floor reconstruction is required [34,38]. Additionally, if the medial buttress is resected, it can lead to alar collapse and eventual nasal obstruction and cosmetic deformity. Utilization of bony-free flap reconstruction can help to address this concern and possible complication depending on the extent of resection.

In cases of orbital exenteration, a large free flap is needed to obliterate both the orbit and maxillary sinus. In these cases, an anterolateral thigh, rectus abdominus, or latissimus-free flap is useful due to the larger bulk available from these donor sites [38]. Osteocutaneous radial forearm- or fibula-free flaps are alternative options to provide a bony reconstruction of the palate to support future dental rehabilitation. The decision is dependent on the patient’s premorbid dentition, the extent of remaining alveolus after the tumor resection, and the size of the defect.

In a single-center retrospective review over 15 years, Cordeiro and Chen described free tissue transfer, particularly rectus abdominis and radial forearm donor sites, as an effective strategy for complex maxillary reconstruction [34]. They also propose an algorithm for reconstruction based on the extent of maxillectomy, involvement of the orbital floor and palate, and status of the orbital contents [34]. In a recent retrospective review analyzing postoperative complaints in 58 patients who underwent maxillectomy and free flap reconstruction, sinonasal complaints were common [39]. These included nasal crusting, nasal obstruction, rhinorrhea, facial pain and pressure, and foul odor, and the presence of sinonasal complaints was higher in patients who underwent radiation, although it was not statistically significant in the study [39]. This emphasized the need for careful postoperative monitoring in these patients, especially for patients undergoing radiation therapy, as they may require additional surgery for debulking or endoscopic sinus surgery.

## 5. Ethmoid Sinuses

Reconstruction after malignancies based in the ethmoid sinuses requires consideration of both the anterior skull base and orbit. Skull base reconstruction aims to close CSF leaks and decrease the risk of meningitis. Anterior fossa leaks are typically lower pressure and can vary substantially in size (REF). The resection of smaller tumors with little intracranial extension may result in 1–2 cm unilateral defects, which can be reconstructive with grafts alone. Larger defects of the anterior fossa can extend from the tuberculum sella to the posterior table of the frontal sinus and span between both orbits. These larger defects require vascularized, multi-layer reconstruction. The nasoseptal flap is a mainstay of endoscopic skull base surgery and can be used in this location, and the use of the extended NSF can help with larger defects [18]. Multi-layer reconstruction is also critical, and various techniques are utilized with collagen inlay, fascia, or alloplastic materials plus the vascularized layer [8,16,40,41]. Multiple layers in the reconstruction have been shown to be the most important variable, with the exact choice of material and inlay/onlay technique being less salient [8,41].

Reconstruction after the resection of ethmoid sinus malignancies also requires consideration of the orbit. In our experience, if the periorbita is not violated, no definitive reconstruction is indicated. However, if the periorbita is violated or resected, consideration must turn to the status of the orbital floor. If the periorbital is no longer present and the infromedial strut is removed, orbital floor reconstruction should be considered as described in the above section on maxilla reconstruction. Other authors have described reconstruction with split calvarial bone grafts, fascia lata, pericranium, and mesh implants after the resection of the lamina paprycea and periorbita [22].

## 6. Frontal Sinus

Primary frontal sinus malignancies are rare but can involve the anterior or posterior table and orbit. The reconstruction of the frontal sinuses follows the above discussion in preventing CSF leaks and maintaining orbital function. Frontal sinus malignancies involving the skull base require careful reconstruction to close any CSF leaks and prevent meningitis. Similar to anterior skull base reconstruction after the resection of ethmoid sinus tumors, free mucosal grafts can be used for small defects without significant intraoperative CSF leaks. Additionally, the nasoseptal flap can also be used in this setting but may not have an appropriate length to reach past the crista galli with the pedicle at the sphenopalatine foramen. For this reason, the extracranial pericranial flap, with the pedicle of the supraorbital and supratrochlear arteries, is heavily utilized for reconstructions in this area [24,25,41]. Fascia lata and temporalis fascia grafts have also been described for anterior skull base dural reconstruction but are not vascularized grafts [22].

Additionally, it is important to consider the preservation of the frontal recess patency. This involves ensuring a patent frontal sinus outflow tract to prevent mucoceles and infection. This may require subsequent endoscopic sinus surgery after healing from the initial resection to restore the outflow tract.

## 7. Sphenoid Sinus

Reconstruction after sphenoid sinus surgery has been heavily discussed in the context of benign pituitary masses and primary skull base tumors, like meningioma and craniopharyngioma [1]. The reconstruction of this area focuses on closing CSF leaks and protecting the carotid artery if exposed. The reconstruction of this area in the setting of malignancy can be complicated by additional dissection into the cavernous sinus to resect areas of perineural spread along the maxillary division of the trigeminal nerve via the resection of the lateral wall of the sphenoid sinus.

Free mucosal grafts can be used for small defects or part of a multi-layer closure. A recent retrospective review of 485 patients between two institutions proposed an algorithm for reconstruction after pituitary surgery based on CSF leak grade [42]. They found that grade 1 CSF leaks in sellar or parasellar reconstructions were adequately reconstructed with free mucosal grafts and that vascularized grafts are appropriate for grade 2 and 3 CSF leaks and large defects [42,43]. However, the nasoseptal flap is heavily used for reconstruction after transsphenoidal surgery due to its reliability and significant impact on decreasing postoperative CSF leaks. This is an option with reliable anatomy and success. Similar to reconstruction after the resection of frontal sinus malignancies, pericranial flaps can be useful in reconstruction after sphenoid mass resection given the reliable anatomy and large available size, especially if a nasoseptal flap is unavailable due to the malignancy or prior surgery [24].

## 8. Clivus

Clival reconstruction follows similar principles to other sinonasal subsite reconstruction, but it is unique as it commonly is associated with a large defect and high-flow CSF leaks. The prepontine cistern is also under higher pressure, so the closure of CSF leaks in this area is associated with higher rates of failure and the more liberal use of lumbar drains. Multi-layer closure is essential for the repair of these defects. Several authors describe an inlay graft, onlay graft, autologous fat, and a vascularized flap [9,44]. The autologous fat is useful to fill the bony defect to prevent pontine herniation [9,44,45]. This is important to consider because pontine herniation or encephalocele can change the postoperative radiation field or increase the risk of radiation to the brainstem [45]. Nasoseptal flaps are a good option for a vascularized graft in the multi-layer closure of clival defects but often need to be extended along the nasal floor to the sidewall to harvest an appropriately wide flap for reconstruction [9]. Inferior turbinate flaps have also been described for the repair of clival defects [27,46]. They have been particularly described by Choby et al. for the repair of small midclival defects [46]. If a nasoseptal or another local flap is not available or not large enough, a TPF or pericranial flap can be used [27]. The pericranial flap can also be used in the context of revision surgery or persistent CSF leaks, as described in a series by Gode et al. [47].

Prior radiation therapy is a particular issue in clival reconstruction because the resection of a clival mass is often performed in a salvage setting after failed radiation or recurrence. This is problematic because the local tissues may not have adequate blood supply for vascularized grafts after radiation therapy. This leads to an increased use of free tissue transfer for clival reconstruction compared to reconstruction at other subsites [48,49]. Radial forearm-free flaps, in particular, have been described as a feasible donor site for clival reconstruction. For posterior fossa defects, the facial vessels can still be used in a transmaxillary fashion; however, a posterior tunnel can also be created from the neck working from underneath the posterior belly of the digastric through the parapharyngeal space. Lastly, for any patient with a CSF leak in the posterior fossa, consideration should be given to using a postoperative lumbar drain.

## 9. Conclusions

Reconstruction after sinonasal malignancy is complex and highly dependent on the subsites involved in the resection. The goals of reconstruction aim to close any CSF leak, restore paranasal sinus function, provide a nasal airway, and optimize quality of life. The nasoseptal flap is a mainstay of reconstruction for these sites, but other locoregional vascularized flaps are available, in addition to free tissue transfer.

## Figures and Tables

**Figure 1 cancers-16-00242-f001:**
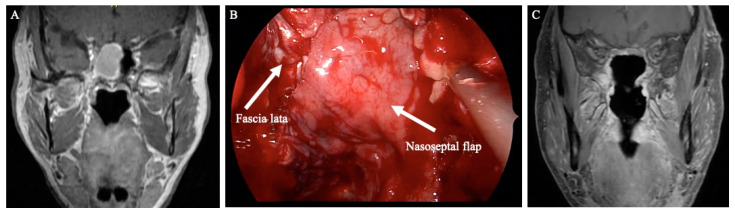
(**A**) Preoperative T1 post-contrast coronal image demonstrating a large right sinonasal mass. (**B**) Intraoperative endoscopic view of the anterior skull base after esthesioneuroblastoma resection. The skull base was reconstructed with a synthetic dura, fascia lata, and a left-sided pedicled nasoseptal flap. (**C**) Postoperative coronal MRI image demonstrating skull base reconstruction along the resection bed.

**Figure 2 cancers-16-00242-f002:**
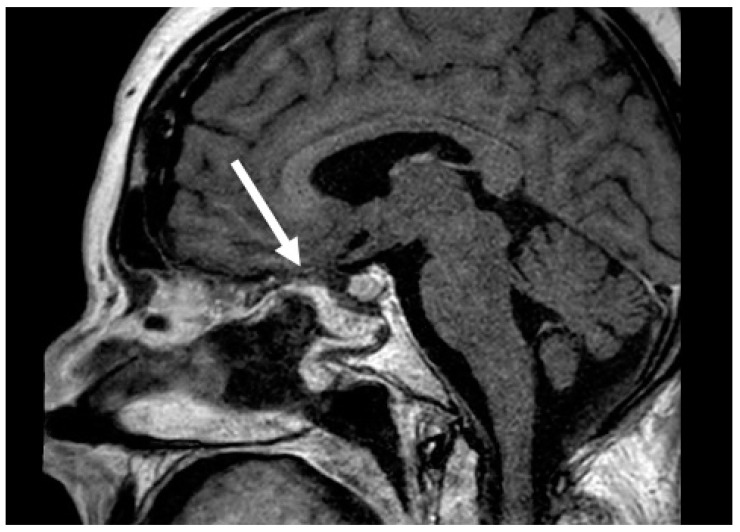
Postoperative T1 post-contrast sagittal MRI images of a well-healed nasoseptal flap (arrow).

**Figure 3 cancers-16-00242-f003:**
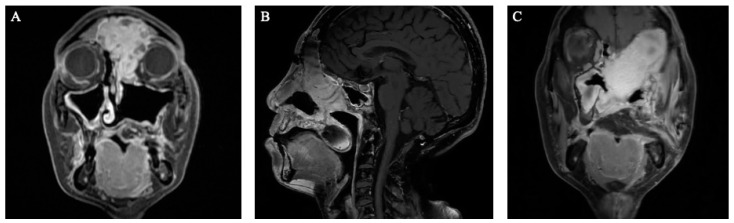
A patient with recurrent adenoid cystic carcinoma. (**A**) Preoperative coronal MRI image demonstrating extensive orbital and skull base involvement. (**B**) Postoperative post-contrast T1-weighted sagittal MRI images after eft orbital exenteration, suprastructure maxillectomy, frontal craniectomy, and ALT reconstruction (**C**) Postoperative post-contrast T1-weighted coronal MRI images.

**Table 1 cancers-16-00242-t001:** Summary of reconstructive options available for skull base reconstruction after sinonasal malignancy resection.

Graft Types	Donor Site	Uses
Non-vascularized	Middle turbinate	Low-flow CSF leaks, small dural defects, and component of multi-layer closure
	Nasal floor	
	Fascia lata	
	Temporalis fascia	
Pedicled	Nasoseptal flap	High-flow CSF leaks and large dural defects
	Pericranial flap	
	Tempoparietal fascia	
	Inferior turbinate	
	Lateral nasal wall	
Free tissue transfer	Radial forearm	High-flow CSF leaks, large dural defects, and paranasal sinus obliteration
	Anterolateral thigh	
	Rectus abdominus	
	Serratus anterior	
	Latissimus dorsi	

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
