# Peer review of "Skull Base Reconstruction by Subsite after Sinonasal Malignancy Resection"

_cancers, 2024, doi:10.3390/cancers16020242_

Round 1

Reviewer 1 Report

Comments and Suggestions for Authors

Good job! High scientific content of the article

Author Response

Thank you for your time and review of our article. 

Reviewer 2 Report

Comments and Suggestions for Authors

The present study/review about skull base reconstruction in sinonasal malignancies management although well written and conducted unfortunately does not add neither new knowledge nor new management strategies to the exisiting body of literature.

I do believe that the study could be greatly improved if the authors would perform a PRISMA review, would shorten all sections that are too long and often redundant and finally develop a clinical algorythm to guide pratictioners  in their routine practice to apply the best reconstruction method/technique facing different situations.

Author Response

Thank you for the review. We have added illustrative cases to help aid to help highlight the text and serve as an aid to readers in applying the reconstruction concepts presented in this review. 

Reviewer 3 Report

Comments and Suggestions for Authors

In this manuscript the authors report a review of skull base reconstruction techniques after sinonasal malignancy resection.

A table could be added in order to schematize the various skull base reconstruction options.

The authors could evaluate to add subheadings in order to improve the readability of the text.

Some illustrative cases could also be added (with radiologic images, as well as intraoperative pics and videos). I believe this would increase the value of the manuscript.

Author Response

Thank you for your review. We have added a table to help with a general overview and characterization of various reconstructive options. We also added subheadings throughout the text to aid in readability. Finally, we also added figures of intraoperative photos, preoperative, and postoperative MRI images.

Round 2

Reviewer 3 Report

Comments and Suggestions for Authors

The authors replied to my comments, thanks.

I would suggest adding pre-, intra-, and post-operative images for the different cases in order to better illustrate the extensions of the lesions that were removed, the intraoperative resection and reconstruction, and the post-operative results.

Author Response

Thank you for your review. We have added preoperative and postoperative MRI images to the figure with the intraoperative. Unfortunately, intraoperative images associated with the other cases are unavailable.